# Fabrication of Humidity-Resistant Optical Fiber Sensor for Ammonia Sensing Using Diazo Resin-Photocrosslinked Films with a Porphyrin-Polystyrene Binary Mixture

**DOI:** 10.3390/s21186176

**Published:** 2021-09-15

**Authors:** Soad Ahmed, Yeawon Park, Hirofumi Okuda, Shoichiro Ono, Sergiy Korposh, Seung-Woo Lee

**Affiliations:** 1Graduate School of Environmental Engineering, The University of Kitakyushu, 1-1 Hibikino, Kitakyushu 808-0135, Japan; z8daa401@eng.kitakyu-u.ac.jp (S.A.); b0daa001@eng.kitakyu-u.ac.jp (Y.P.); okuda-hirofumi@toyoko-jp.com (H.O.); a9maa006@eng.kitakyu-u.ac.jp (S.O.); 2Department of Electrical and Electronic Engineering, University of Nottingham, Nottingham NG7 2RD, UK; s.korposh@nottingham.ac.uk

**Keywords:** ammonia detection, layer-by-layer, U-bent optical fiber, porphyrin, poly(styrene sulfonate), diazo resin, photocrosslinking

## Abstract

Ammonia gas sensors were fabricated via layer-by-layer (LbL) deposition of diazo resin (DAR) and a binary mixture of tetrakis(4-sulfophenyl)porphine (TSPP) and poly(styrene sulfonate) (PSS) onto the core of a multimode U-bent optical fiber. The penetration of light transferred into the evanescent field was enhanced by stripping the polymer cladding and coating the fiber core. The electrostatic interaction between the diazonium ion in DAR and the sulfonate residues in TSPP and PSS was converted into covalent bonds using UV irradiation. The photoreaction between the layers was confirmed by UV-vis and Fourier transform infrared spectroscopy. The sensitivity of the optical fiber sensors to ammonia was linear when exposed to ammonia gases generated from aqueous ammonia solutions at a concentration of approximately 17 parts per million (ppm). This linearity extended up to 50 ppm when the exposure time (30 s) was shortened. The response and recovery times were reduced to 30 s with a 5-cycle DAR/TSPP+PSS (as a mixture of 1 mM TSPP and 0.025 wt% PSS in water) film sensor. The limit of detection (LOD) of the optimized sensor was estimated to be 0.31 ppm for ammonia in solution, corresponding to approximately 0.03 ppm of ammonia gas. It is hypothesized that the presence of the hydrophobic moiety of PSS in the matrix suppressed the effects of humidity on the sensor response. The sensor response was stable and reproducible over seven days. The PSS-containing U-bent fiber sensor also showed superior sensitivity to ammonia when examined alongside amine and non-amine analytes.

## 1. Introduction

The use of optical fibers for chemical and biological sensing has become popular for both scientific and commercial applications [1,2]. Because of their unique properties, such as small size, immunity to electromagnetic interference, and biocompatibility, they facilitate remote and real-time measurements with high sensitivity and selectivity. Most importantly, an optical fiber sensor (OFS) can be deployed in a harsh corrosive environment owing to the chemically inert silica substrate of the fiber [3].

The different OFS geometries reported in the literature for refractive index (RI) and absorption sensings include straight decladded [4], partially polished [5], laterally polished [6], D-shaped [7], and U-bent [8] fibers. Among the various reported designs, the U-bent design has several advantages, such as (1) a high evanescent wave absorbance sensitivity due to conversion of the lower-order modes into higher-order modes, (2) less fragility compared to other geometries, (3) ease of probe fabrication and repeatability, and (4) ease of development into a point sensor. Combining the advantages of the U-bent geometry and the stripped fiber core functionalized with nanomaterials enables the development of a high-sensitivity sensor capable of binding target chemical species.

Ammonia (CAS No. 7664-41-7) is a widely used chemical in the chemical industry. It is an important gas for assessing indoor air quality, particularly in the industrial sector, as its presence in excess of its exposure limit results in health issues [9]. Its permissible exposure limit is regulated to be 50 ppm (35 mg m^3^), according to the Occupational Safety and Health Administration (OSHA) guidelines [10]. On the other hand, ammonia has recently attracted a great deal of attention as a hydrogen carrier and an alternative energy resource to replace hydrogen [11]. From this perspective, the development of technologies related to ammonia gas detection, separation, condensation, etc., becomes an important issue. Moreover, the use of ammonia as a biomarker in human breath [12,13] for detecting pathological disorders such as renal insufficiency [14], hepatic dysfunction [15], *Helicobacter pylori* infection [16], and halitosis [17], has recently gained significant interest from researchers [18].

From this perspective, OFSs provide an excellent base for developing low-cost, small, sensitive, and reliable ammonia sensors, making them viable for on-field applications. In the past decades, sensing agents made of various organic polymers, dyes, and pH indicators using OFSs have been investigated for detecting ammonia. In these cases, the ammonia reacts with the water molecules to produce hydroxide ions (OH^−^), which deprotonate the sensing agent, resulting in a change in the absorption spectrum of the film [19]. 

In our previous work, we demonstrated evanescent-wave-based OFSs for ammonia gas sensing, which were modified with tetrakis(4-sulfophenyl)porphine (TSPP) and several cationic polymers via electrostatic interactions between the oppositely charged polyelectrolytes [20,21,22]. The exposure of the TSPP nanonassembled film to ammonia induced unique optical changes in the transmission spectrum of the optical fiber, displaying the characteristic absorption bands (two Soret bands and Q bands) of the assembled TSPP compound [20]. A high extinction coefficient (>2.0 × 10^5^ cm^−1^ M^−1^) makes porphyrin particularly viable for the creation of optical sensors. However, several issues challenge its implementation when used with optical fibers, such as leaching out of the TSPP from the nanoassembled films [21], undesirable sensitivity to humidity owing to the hydrophilicity of the porphyrin, swelling of the immobilized polymer, long recovery time at high ammonia concentrations (>1 ppm), and a longer requisite exposure time (>30 s) [22].

A solution to such problems is creating covalent bonds or effective interactions between the dyes and the embedding matrix. It has been reported that polyelectrolyte complexes that are formed using diazo resin (DAR) as a cationic polyelectrolyte can be converted into covalent bonds by irradiation with ultraviolet (UV) light [23]. Crosslinked DAR films are stable even at high pH due to their high resistance to solvent etching, resulting in denser and more rigid films [24].

The present study proposes a novel strategy for fabricating a U-bent OFS coated with covalently crosslinked porphyrin layers. As mentioned above, the U-bent fiber geometry is advantageous in improved evanescent wave interactions due to the ergonomic design of the fiber compared to other fiber geometries. DAR as a polycation polymer and a binary mixture of TSPP and poly(styrene sulfonate) (PSS) as polyanions were alternately deposited as a film by the layer-by-layer (LbL) method on the U-bent fiber core. It is presumed that the presence of PSS in the matrix imparts greater stability to the low-molecular-weight dyes or pigments in the film and reduces the interference due to water on the sensor response. On the other hand, it may affect the mobility or aggregate structures of the TSPP, and thus, the amount of PSS employed in the film was optimized. Photopolymerization via UV irradiation resulted in a greater stability of the TSPP inside the film, which was co-assembled with PSS. Herein, the responses of the sensor to different concentrations of ammonia were measured under nearly saturated humidity conditions, and the critical sensor parameters, such as selectivity, sensitivity, and stability, were investigated.

## 2. Materials and Methods

### 2.1. Materials

TSPP (Mw: 934.99 g mol^−1^) was purchased from Tokyo Chemical Industry Co., Ltd. (Tokyo, Japan). PSS (Mw: 70.000 g mol^−1^) and acetone were purchased from Sigma-Aldrich (St. Louis, MO, USA). NH_4_OH (28 wt% in H_2_O), which was used to produce the ammonia analyte gas, and pyridine (Py), ethanol, toluene, methanol, and 1-hexanol, which were used as additional analyte gases, were purchased from Wako Pure Chemical Industries (Osaka, Japan). Trimethylamine (TMA, 30% in H_2_O) and 1-propanol were purchased from Kanto Chemical Co., Ltd. (Osaka, Japan). Triethylamine (TEA) was purchased from Kishida Chemical Co., Ltd. (Osaka, Japan). All chemicals were of analytical grade and were used without further purification. DAR (Mw: ca. 2500 g mol^−1^) was synthesized according to a previously reported method [25,26]. Deionized (DI) pure water (18.2 MΩ·cm) was obtained by reverse osmosis, followed by ion exchange and filtration (Aquapuri 541; Young In Scientific, Seoul, Korea).

### 2.2. Preparation of Optical Fiber Sensor

To fabricate the OFS, a short section of the plastic cladding of a multimode optical fiber (HWF 200/230/500T silica core/plastic cladding with a 200 μm core diameter, CeramOptec: Bonn, Germany) was burned and bent to a U-shape using a burner. The total length of the fabricated U-bent OFS was approximately 40 cm, and the exposed length of the U-shaped tip probe (without plastic cladding) was approximately 1 cm, as shown in Figure 1a. Then, the exposed section of the silica core was rinsed in ethanol and DI water several times, and treated with 1 wt% ethanolic KOH (ethanol/water = 3:2, *v*/*v*) for 20 min. Thus, the negatively charged evanescent region was formed.

Next, an electrostatic film was deposited by alternately immersing the negatively charged activated core of the optical fiber into an aqueous DAR solution (1 wt%, pH 2.2) and a mixture of TSPP (1 mM in H_2_O, pH 3) and PSS (0, 0, 0.025, and 0.1 wt% in H_2_O) at room temperature for 15 min each, producing a DAR/TSPP+PSS bilayer (where one cycle results in a DAR/TSPP or DAR/TSPP+PSS bilayer). This was achieved by adding the coating solution (volume 70 μL) to a deposition cell with the intermediate processes of water washing and drying by flushing with nitrogen gas being. Multilayer assemblies can be formed by cyclic repetition of these two steps. Figure 1b shows the LbL assembly of the photocrosslinkable DAR and TSPP or a binary mixture of TSPP and PSS. An LbL film without PSS was prepared to evaluate the effect of PSS on the sensor response, and the outermost layer of the alternate film was DAR in all cases. The films were deposited in the dark to avoid the decomposition of the DAR. Finally, the U-bent optical fiber modified with a 5-cycle DAR/TSPP or DAR/TSPP+PSS thin film was cured by exposing it to UV light (λ_max_ = 365 nm) at a distance of 10 cm for 15 min. Consequently, the ionic bonds between the diazonium cationic ions in DAR and the sulfonate anionic residues in TSPP and PSS can be converted into crosslinked covalent bonds by UV light irradiation.

### 2.3. Experimental Setup and Sensing Performance

The stripped section of the optical fiber that was coated with a functional film was fixed in a specially designed cylindrical acrylic gas measurement chamber (3.6 cm in diameter, 1.0 cm in height; internal volume: 10.2 cm^3^). As seen in Figure 1c, one end of the optical fiber was connected to a deuterium halogen light source (DH-2000-Ball, Mikropack), while the other end was connected to a spectrometer (S1024DW, Ocean Optics) to monitor the assembly process and gas sensing. Spectra Suite^®^ Spectrometer Operating Software (Ocean Optics) was used for the analysis. The absorbance (A) was determined by taking the logarithm of the ratio of the transmission spectrum of the coated fiber, T(λ), to the transmission spectrum measured prior to the film deposition, T_0_(λ), as shown in Equation (1):(1)$$A\left(\lambda \right)=-\log\;[\frac{T\left(\lambda \right)}{T_{0}\left(\lambda \right)}].$$

The response of the U-bent OFS was measured by exposing it to ammonia and other analyte gases at different concentrations, which were generated by placing a fixed volume (100 μL) of their aqueous analyte solutions of different concentrations inside the measurement chamber near the U-bent tip. Each analyte solution was kept for approximately 3 min to attain equilibrium between the released gas and the solution (also kept until the saturation of the signals measured at the wavelength of 706 nm). The individual amine gas concentrations of ammonia, TMA, TEA, and Py were measured using gas tubes (No. 3L for ammonia, No. 180 for TMA, No. 180L for TEA, and No. 182 for Py; GasTech, Inc., Ayase, Japan). The actual amine gas concentrations were estimated using calibration curves obtained from the corresponding amine concentrations in the solutions, as shown in Appendix A.

The baseline spectrum of each experiment was measured by placing DI water (100 μL) at the bottom of the measurement chamber near the tip of the U-bent optical fiber until the signals measured at the wavelength of 706 nm stabilized. The relative humidity level inside the measurement chamber during all measurements reached approximately 80% at room temperature (approximately 24 °C) (Appendix A). The sensor response (SR) was calculated using the following equation Equation (2): (2)$$\mathrm{SR}\;(\%)=\frac{I_{0}-I}{I_{0}}\times 100,$$
where *I*_0_ and *I* are the light intensities of the OFS in the absence and presence of an analyte gas, respectively, measured at the same wavelength.

### 2.4. Film Characterization

Scanning electron microscopy (SEM) measurements were performed with a Hitachi S-5200 at an acceleration voltage of 10–15 kV. A 2-nm thick platinum layer was deposited on all samples using a Hitachi E-1030 ion sputter at 15 mA and 10 Pa prior to measurements, preventing the charge up of the sample surface. For Fourier transform infrared spectroscopy (FT-IR) analysis, the films were prepared on a gold-coated silicon wafer plate and measured using a Spectrum 100 FT-IR spectrometer (Perkin Elmer Japan Co., Ltd., Yokohama, Japan). The contact angles of the samples were measured using a DropMaster 100 (DropMaster Co., Ltd., Niiza, Japan).

## 3. Results and Discussion

### 3.1. Strategy for Sensitive and Reproducible Sensor Fabrication

TSPP, which has two distinct and unique aggregates (J- and H-) along with monomers in nanoassembled films, has been extensively investigated in previous studies, including our trials [21,27,28]. The concentrations of these structures usually change with the degree of TSPP protonation or deprotonation [20,29]. These optical features can be used for the detection of some analytes that are proton-donating or proton-accepting in aqueous or gaseous media [22,27]. However, the TSPP aggregations inside the film are strongly affected by humidity because of the hydrophilicity of TSPP. As a result, the water molecules trapped in the film matrix retain the analyte for an extended period. This drawback of the nanoassembled TSPP films for chemical sensing may be caused by the fundamental features of the layered film, which are based on the electrostatic interaction between the alternate layers.

DAR is a candidate molecule for solving this problem because it is capable of changing the electrostatic bond between the layers to a covalent bond. The arene diazonium salt (DAR-N2+) ion-complexed with an anionic species can be easily decomposed via UV irradiation or heat treatment by releasing the N2 gas. Consequently, the negatively charged sulfonate groups of TSPP can be covalently crosslinked through nucleophilic addition to the carbocation formed in the terminal benzene ring of the DAR. Sun et al. reported that negatively charged TSPP sulfonate ions complexed with DAR can be converted into covalent bonds after photoreaction induced by UV irradiation [30]. In this study, a novel approach was demonstrated to improve the hydrophobicity of the sensor films by using a binary mixture of TSPP and PSS. PSS is an anionic polymer having sulfonate groups and makes strong ion-complexes with diazonium functional groups. As illustrated in Figure 1b, the assembled layers can be covalently crosslinked after UV irradiation and become much more hydrophobic due to the neutrally transformed phenyl rings in the PSS. However, the content of PSS employed in the sensor films may competitively affect the sensing ability of TSPP.

### 3.2. Optical Features of Covalently Attached DAR/TSPP+PSS Multilayers

The layered film assembly with DAR and TSPP or DAR and a binary mixture of TSPP and PSS was confirmed after each deposition cycle by monitoring optical changes in the transmission spectra of the optical fiber. Optical fiber absorption spectra were obtained by adapting Equation (1) to the transmission spectra recorded during the film deposition (Appendix A). Figure 2a,b show the absorption spectra of a 5-cycle DAR/TSPP+PSS film with an outermost layer of TSPP+PSS and DAR, respectively, where the PSS concentration in the coating solution was adjusted to 0.025 wt%. This PSS concentration was optimal for improving sensitivity to ammonia and hydrophobicity of the sensor film. Figure 2c shows the absorbance changes at 435, 490, and 706 nm during the film deposition when the outermost layer was deposited with a TSPP+PSS binary mixture. The absorbance at each wavelength increased linearly with the number of layers, indicating the uniform film deposition for each component. The absorption spectra for the individual layers with a TSPP+PSS outermost layer were characterized by a double peak in the Soret band at 435 and 490 nm and a pronounced Q band at approximately 706 nm. In addition to these characteristic peaks, the absorbances at 380 and 309 nm correspond to the π–π* transition of the diazonium group [30] and the benzene rings of DAR and PSS, respectively. Notably, no significant changes in the position or shape of the TSPP absorption peaks were observed when the outermost layer was deposited with DAR, which indicates that the TSPP aggregate structures do not change when overlaid with DAR layers. Moreover, mixing TSPP with a small amount of PSS did not significantly affect the absorption spectra of the film.

The decomposition of the diazonium groups in the 5-bilayer DAR/TSPP+PSS (0.025 wt%) film completed within 15 min. Figure 2d shows a decrease in absorbance at 380 nm, indicating that the diazonium groups decomposed under UV irradiation. Simultaneously, the absorbance of the Soret band of TSPP at approximately 435 nm also slightly decreased without any change in the peak position and shape. This small decrease at 435 nm may be due to masking of the diazonium group spectra. Therefore, we concluded that the absorbance at 435 nm is mainly composed of the Soret band of the TSPP but partly overlaps with the edge of the diazonium group band before UV irradiation. This result also suggests that no significant changes occurred in the film conformation during the photocrosslinking reaction, as reported in a previous study [24]. On the other hand, the Q band of TSPP was blue-shifted (ca. 3 nm) with a slight increase in the absorbance at 706 nm, indicating that UV irradiation may induce structural changes in TSPP J-aggregates.

### 3.3. Sensor Responses to Ammonia

According to the equilibrium equation (Equation (3)), the ammonia gas that evaporated from the aqueous ammonia solutions saturated inside the measurement chamber.
(3)$${\mathrm{NH}}_{4}\mathrm{OH}\left(\mathrm{aq}\right)⇆{\mathrm{NH}}_{3}\left(g\right)+H_{2}O$$

Figure 3a shows the intensity changes (defined as ∆*I*) in the transmission spectra due to the presence of ammonia in the 5-cycle DAR/TSPP+PSS (0.025 wt%) film, which is obtained by subtracting the transmission spectrum measured at a given ammonia concentration from that measured using water. As the ammonia concentration increased from 0 ppm to 100 ppm (sol), the intensity changes in the transmission spectra increased at 485 and 706 nm and decreased at 411 and 518 nm, respectively, as shown in Figure 3b. The most perceptible change in the intensity of the transmission spectra was observed at 706 nm. This result suggests the disappearance of the electrostatic interactions between the TSPP molecules due to the deprotonation process induced by the adsorption of ammonia molecules [22,31], indicating the distortion of the TSPP J-aggregates in the film.

As evident from Figure 3c, the Q band centered at 706 nm shows perceptible absorbance changes in the corresponding absorption spectra obtained using Equation (1), which are similar to the results observed in the transmission spectra shown in Figure 3a. The absorption peaks at 490 and 706 nm, which are attributed to the TSPP J-aggregates in the film, were reduced, while the absorbance of the peak at 435 nm increased.

Figure 3d shows the intensity changes at 706 nm, obtained from the four consecutive ammonia measurements in the concentration range from 0 to 100 ppm (sol). The intensity changes after the first measurement of the as-prepared film exhibited good reproducibility, with a standard deviation of ±4.5 mV at 100 ppm concentration of ammonia, although the average intensity change of ammonia measured thrice at 70 ppm (sol) was approximately 70% of the initial measurement. This decrease in intensity change after the first measurement of the as-prepared film may be due to the optimization of the film structure after continuous use. The higher amount of TSPP J-aggregates is present in the as-prepared film; however, some of the TSPP molecules, particularly those covalently bound to the DAR, are hard to return to the original J-aggregation state after the first use. The film matrix appears not to recover the initial state completely. Consequently, the TSPP J-aggregation is subtly realigned after the initial exposure to ammonia and optimized for ammonia sensing. 

### 3.4. Optimization of the Content of PSS Employed in the Sensor Film

The reasons for employing PSS with TSPP in this study are as follows: first, a PSS polymer has multiple sulfonate functional groups and can be crosslinked with DAR upon UV irradiation, which helps form a covalently bonded stable film [23,24]; second, PSS may enhance the hydrophobicity of the film when added by mixing it with TSPP; the enhanced hydrophobicity of the film reduce the desorption of TSPP from it as well as its affinity to water molecules. These advantages of PSS improve the stability and sensitivity of the crosslinked film.

To investigate the effect of PSS in the sensor film on ammonia gas sensing under highly humid conditions, three types of covalently crosslinked DAR/TSPP+PSS films with different PSS contents of 0 wt%, 0.025 wt%, and 0.1 wt% were prepared. Figure 4a shows an example of the dynamic SRs measured at 706 nm for the U-bent OFS coated with a DAR/TSPP+PSS (0.025%) film when exposed to different ammonia concentrations from 0 to 100 ppm (sol). The SR of the sensor film was recorded for 3 min after placing 100 μL of aqueous ammonia of a given concentration in the chamber; the ammonia solution was then removed by suction and the chamber was flushed with dry air at a flow rate of 1 L min^−1^ for 1 to 2 min. The SR, calculated using Equation (2), was exceedingly quick (<30 s) for each ammonia solution and recovered almost entirely even after exposure to a high concentration of ammonia at 100 ppm (sol), exhibiting a SR of approximately 10%. 

Figure 4b compares the SRs of the three DAR/TSPP+PSS sensor films, which were measured at 706 nm in all cases. Interestingly, the SR was influenced by the PSS content and increased in the following order: PSS 0.1 wt%, PSS 0 wt%, and PSS 0.025 wt%. In particular, the sensor film without PSS exhibited a slightly different adsorption behavior, having two linear ranges (0–7 ppm and 7–17 ppm), compared to the other two sensor films containing PSS, indicating a slow reaction at low ammonia concentrations. It may be that the diffusion of ammonia into the film is disturbed at low concentrations.

### 3.5. Sensitivity and Selectivity of the PSS-Containing Sensor Films

As seen in Figure 4b, the linear behavior of the calibration curves for ammonia were observed within 20 ppm (sol). Interestingly, the film prepared with 0.025 wt% PSS exhibited a high SR compared to the other two films. Except for the sensor film without PSS, the SR of both PSS-containing sensor films to ammonia was almost linear with respect to the ammonia concentration in the range of 0–17 ppm (sol). It is presumed that embedding an appropriate amount of PSS inside the film increases the film’s hydrophobicity and stability after photocrosslinking, suggesting that reduced water condensation inside the film allowed the permeation of analyte gases under highly humid conditions. As a result, the TSPP was able to repeatedly detect the presence of ammonia. Conversely, the rigidity of the film upon PSS reinforcement may hinder ammonia gas diffusion into the film matrix. Consequently, a decrease in sensitivity was observed when doping the TSPP layer with a higher PSS content (>0.1 wt%). Thus, the optimum PSS concentration was found to be 0.025 wt%. Details of the response and recovery times of the fiber sensors are listed in Table 1.

The noise levels of the SRs were calculated from the baselines in a steady state before exposure to ammonia and were used to determine the limit of detection (LOD) of the U-bent fiber sensors for ammonia (data not shown). Three or more stable baselines were selected from the dynamic responses to ammonia in Figure 4a, and a value of 0.032 ± 0.002% SR was estimated as to be the noise value. Therefore, 0.096% SR at a 3 s level was used as the LOD. As described in Figure 4b and Table 1, the best ammonia sensing performance was observed in the DAR/TSPP+PSS (0.025 wt%) film, exhibiting an LOD of 0.23 ppm (sol) in addition to its fast response and recovery times (each 30 s). This LOD value is about three times smaller than that of the sensor film with 0.1 wt% PSS. Interestingly, the linear trend of the sensor response–concentration curve for the sensor with 0.025 wt% PSS was extended up to 50 ppm (sol) of ammonia when the exposure time (30 s) was shortened (Appendix A).

On the other hand, the curve for the sensor film without PSS displayed a shorter linear trend only in the concentration range of 0–7 ppm with an LOD of 1.25 ppm (sol) of ammonia. At a higher concentration range of 7–17 ppm, the film’s response was enhanced, exhibiting a sensitivity of 0.34 ± 0.03% ppm^−1^. This nonlinear sensitivity of the sensor film without PSS is probably because of a different diffusion mechanism of the ammonia into the film compared to that of both PSS-containing sensor films. Water molecules condensed on the surface of the sensor films disturb the diffusion of ammonia into the film owing to the high solubility of ammonia in water. Therefore, the hydrophobicity of PSS embedded in the film contributes to the efficient diffusion of ammonia gas at a high relative humidity.

An important sensor parameter is selectivity, that is, the capability to detect a particular substance from other coexisting substances. Therefore, the U-bent OFS coated with a 5-cycle DAR/TSPP+PSS (0.025 wt%) film was exposed to several amine and non-amine vapors along with ammonia. Figure 5a shows the calibration curves based on the intensity change, ∆*I* as sensor response, measured at 706 nm upon exposure to the vapors of ammonia, TMA, TEA, and Py, which were present individually or as a mixture of equal concentrations in the range of 0–17 ppm (sol). The DAR/TSPP+PSS (0.025 wt%) film exhibited the highest response to ammonia and then to TMA starting at 10 ppm (sol). In contrast, no sensitivity to TEA and Py was observed throughout the measured range (∆*I* at 706 nm < 2 mV). The fiber sensor is most sensitive to ammonia among the four amine analytes; however, the sensitivity to ammonia decreased slightly in a mixture of the four amine analytes. This decrease means that the high sensitivity of the sensor to ammonia cannot be explained only by the basicity of the analytes alone (conjugate acid dissociation constant at logarithmic scale, p*K*a, at 25 °C: TEA, 10.75 > TMA, 9.87 > ammonia, 9.36 > Py, 5.25) [32]. An additional decisive parameter may be the vapor pressure of each analyte, which is shown in decreasing order: 6650 mmHg (21 °C) >> 430 mmHg (21 °C) > 51.75 mmHg (20 °C) ≅ 23.8 mmHg (25 °C for ammonia, TMA, TEA, and Py, respectively [32]. The SR to ammonia was four-fold larger than that to TMA at 15 ppm (sol) (ca. 60 and 15 mV for ammonia and TMA, respectively). In addition, the calibration range for TMA was narrow, ranging from 7 to 20 ppm (sol), compared to the concentration range of 0 to 17 ppm (sol) for ammonia.

Figure 5b shows the calibration curves based on the intensity change at 706 nm of the 5-cycle DAR/TSPP+PSS (0.025 wt%) film for different gas concentrations of ammonia and TMA. The actual gas concentrations of the four amine analytes (ammonia, TMA, TEA, and Py) vaporized from their corresponding aqueous solutions were measured using gas tubes (Appendix A). The linear correlation coefficients between the liquid and gas phase concentrations were estimated to be 0.090, 0.225, 0.059, and 0.188 for ammonia, TMA, TEA, and Py, respectively. It is presumed that the actual gas concentrations of TMA and Py are much higher than those of ammonia in the given concentration range. However, TMA was only detectable at gas concentrations over 1.5 ppm. Regardless of the relatively high gas concentrations of TMA and Py, the U-bent OFS surprisingly revealed the highest sensitivity to ammonia. Therefore, the decreased sensitivity in the case of the mixture of all four amines may be attributed to the interference from the less sensitive TMA and Py with relatively high gas concentrations.

Details of the sensitivity and LOD of the 5-cycle DAR/TSPP+PSS (0.025 wt%) film obtained when the solution and gas concentrations for ammonia and TMA were used are summarized in Table 2. When the actual gas concentrations for ammonia and TMA were used for sensitivity calculation, the LOD for ammonia was estimated to be about 0.03 ppm from the calibration curve in Figure 5b for the DAR/TSPP+PSS (0.025 wt%) film, whereas the LOD for TMA was 1.65 ppm, which is about 60 times the LOD for ammonia. From the calibration curves in Figure 5b, it is supposed that about 0.1 ppm of ammonia may be detectable even when 2 ppm of TMA coexists in gas samples. In practice, the LOD for the amine mixture is approximately ten times larger than that for ammonia, indirectly representing that the coexisting amines interfere slightly with the ammonia sensor response.

On the other hand, this LOD (2.85 × 10^−2^ ppm) for ammonia is surprisingly 100 times smaller than that of the same 5-cycle PDDA/TSPP film, where PDDA is poly(diallyldimethylammonium chloride, deposited on a linear optical fiber, which was reported in a previous study [22]. In addition, when considering the molecular weights of ammonia (17.03 g mol^−1^) and TMA (59.11 g mol^−1^), the LOD for ammonia was about 17 times smaller than that for TMA in the gas phase, which implies a higher selectivity of the sensor to ammonia.

Figure 6a,b show the dynamic responses of the intensity changes and the intensity change-based relative SRs measured at 706 nm for ammonia, amines, and non-amine analytes. Non-amines such as hexanol, ethanol, propanol, acetone, and methanol exhibited slightly decreased intensity changes compared to those of ammonia and other amines. It was confirmed that the transmission spectra of the U-bent fiber sensor were influenced over the entire spectral range (Appendix A). These changes in the transmission spectra may be attributed to increased light scattering due to the RI values of the non-amine analytes [32], because the order of RI for the non-amine analytes is hexanol (1.4178) > propanol (1.377) > acetone (1.36) ≈ ethanol (1.36) > methanol (1.33) ≈ water (1.33). However, the effects of the RI cannot be generalized, such as in the cases of toluene (1.50) and Py (1.51), where the intensity remained unchanged at less than 2 mV over the tested concentration range of ~10,000 and 100 ppm, respectively, as shown in Appendix A. Additionally, the film structure may be another reason. The film includes phenyl and secondary amine moieties, which increase the optical thickness and result in a decrease in the intensity over the entire spectral range when the film is exposed to alcoholic and polar analytes. Therefore, we conclude that the DAR/TSPP film with PSS is specifically sensitive and size-selective to ammonia among the low-molecular-weight volatile amine analytes through acid–base interactions.

### 3.6. Stability and Reproducibility of the Sensing System

To confirm the long-term stability of the SR, the DAR/TSPP+PSS (0.025 wt%) film was repeatedly exposed to 7 ppm (sol) ammonia over seven days. As shown in Figure 7a, the ammonia SR was almost stable over seven days, showing an average intensity change of 34.8 ± 1.5 mV, which is approximately 77% of the first measurement result of the as-prepared film, where the standard deviation of ±1.5 mV represents a variation of approximately 0.4 ppm in ammonia concentration from the sensitivity value (4.05 mV ppm^−1^) in Table 2. As mentioned above, this small decrease in intensity at 706 nm may be due to the optimal placement of non-crosslinked TSPP molecules inside the film. In addition, similar results were obtained in a series of repeated ammonia exposure experiments over a short period of time (inset in Figure 7a).

The stability and reproducibility of the sensor response was also confirmed when the sensor film was exposed to 50 ppm (sol) ammonia (Figure 7b). Improvements in sensor response and recovery times were observed in the PSS-embedded film when a 50 ppm (sol) ammonia was repeatedly applied to both sensor films with and without PSS (0.025%) (Appendix A). A twice as long exposure duration (>120 s) was required for the sensor film without PSS compared to that (ca.60 s) of the film containing PSS. Additionally, a much longer time was required for the recovery of the sensor response. One interesting finding was that short-time exposure (<30 s) to ammonia gas resulted in high stability and quick recovery of the sensor response, as reported in a previous study [33]. Herein, the sensor response to a relatively high ammonia concentration (50 ppm) even for a long exposure duration was fully reversible and repeatable over the entire spectral range, as shown in Figure 7b, which was achieved by flushing the film with dry air for a few seconds.

### 3.7. Film Structure and Morphology 

Figure 8 shows the SEM images of the DAR/TSPP and DAR/TSPP PSS (0.025 wt%) sensor films before (Figure 8a,c) and after (Figure 8b,d) ammonia gas exposure. Both films showed rod-like unique structures before gas exposure. Particularly, the DAR/TSPP+PSS (0.025 wt%) film exhibited more extended aggregates and more uniform particles (Figure 8c). However, the film morphologies drastically changed and most of the aggregated structures disappeared after exposure to ammonia gas. On comparing the magnified SEM images in Figure 8e,f, dozens of holes were observed after ammonia gas exposure in both sensor films, particularly in the film without PSS. These differences in film morphology upon the introduction of a small amount of PSS result in the high-performance ammonia gas sensing ability of the DAR/TSPP+PSS (0.025 wt%) film. As morphological changes occur in the film after gas exposure, the light-absorbing and scattering abilities of the exposed film vary compared to those without gaseous exposure, resulting in a change in the evanescent wave and output light intensities of the U-bent OFS.

The formation of covalent linkages following the decomposition of the diazonium group was further verified by FT-IR measurements. Two absorption peaks observed at 2168 and 2222 cm^−1^, which originate from the symmetric and asymmetric stretching vibrational modes of the diazonium ion (N_2_^+^), respectively [23], disappeared entirely after UV irradiation, indicating the decomposition of the diazonium groups. Details of other FT-IR absorption peaks are shown in Appendix A.

### 3.8. Sensing Mechanism

The exposure of the sensor films to ammonia gas deprotonated the J-aggregated TSPP and transformed it to its monomeric state. This deprotonation process distorts and reduces the TSPP J-aggregates, decreasing the absorbance of the film at 706 nm, thereby causing an increase in the absorbance at 435 nm, which indicates the partial transformation to the mono-acidic state. The mono-acidic TSPP molecules were protonated again through ammonia removal from the film after flushing with dry air. As a result, the TSPP J-aggregates were reformed inside the film, regenerating the baseline.

The sensor fabrication and sensing mechanism of the photocrosslinked DAR/TSPP sensor films with and without PSS are shown in Figure 9. The coexistence of PSS in DAR/TSPP layered films can improve the sensitivity and reproducibility of the sensor response owing to the following two factors. First, the enhanced electrostatic interaction between the DAR and PSS layers increases the number of free TSPP molecules. This facilitates reversible conversion between their deprotonated and protonated forms. Second, there was less condensation on the films. The water molecules adsorbed on the film facilitate the formation of a long residue of ammonium ions (NH_4_^+^) produced by the deprotonation of TSPP, requiring a longer recovery time for the sensor response. However, as illustrated in Figure 9b–d, film formation with excess PSS prevents the introduction of TSPP J-aggregates in the film, resulting in a reduced number of ammonia binding sites. The further enhanced film rigidity also reduces the flexibility of the TSPP aggregates inside the film and impedes gas diffusion into the film matrix.

The contact angles of the sensor films may help understand the changes in hydrophobicity due to the PSS content employed in the films. Contrary to our expectations, all sensor films showed similar contact angles of over 80° after photocrosslinking by UV irradiation. Compared to the low contact angles of typical layered films prepared via electrostatic self-assembly [34], the DAR-based covalently crosslinked sensor films have higher hydrophobicity. Interestingly, the contact angle of the DAR/TSPP+PSS (0.025 wt%) film changed from 85.8° to 87.6° in the presence of 70 ppm (sol) ammonia after pure water, despite the decrease in the contact angle in the other two films (0 wt% and 0.1 wt%). This result suggests that adding an appropriate amount of PSS (e.g., 0.025 wt% for 1 mM TSPP in water) encouraged rapid gas diffusion due to the improved hydrophobicity inside the film, resulting in momentary ammonia gas condensation at the solid–gas interface.

As a summary, based on the above results, the key features of the U-bent OFS proposed in this study are compared with the previously reported LBL OFSs for ammonia sensing, as listed in Table 3. The PSS-assisted proposed sensor fabrication has the following advantages: (i) long-term stability over seven days, owing to the covalently crosslinked LBL film attachment, (ii) less leakage or detachment of the indicator (TSPP) in a wet atmosphere, (iii) improved hydrophobicity inside the film structure, (iv) efficient transport of the ammonia analyte to the sensing layer, and (v) enhanced evanescent wave due to the U-bent optical fiber geometry. To date, to the best of our knowledge, there is no sensor with satisfactory sensitivity and selectivity for the detection of ammonia from highly humidified samples.

## 4. Conclusions

Photocrosslinking of DAR and TSPP or DAR and a binary mixture of TSPP and PSS on the core of a U-bent optical fiber was demonstrated for the fabrication of a humidity-resistant OFS. The use of PSS in the film helped obtain stable sensor responses with good gas diffusion into the film. However, the addition of excess PSS inhibited the flexibility of the film and was not beneficial for ammonia gas sensing. Thus, in this study, the optimum PSS concentration was 0.025 wt% with 1 mM TSPP in water. The optimized U-bent OFS exhibited optical responses that had a linear relationship with the aqueous ammonia concentration in the range of 0–17 ppm. The effective response time was less than 30 s, and the baseline was quickly regenerated by flushing the film with dry air after each test. The current sensor system enables the detection of ammonia gas at approximately 30 ppb (parts per billion) concentration. So far, a few standard methods have been developed for ammonia gas sensing, mainly focused on solid-state or electrochemical sensors, along with limited optical sensors. More advanced methods such as gas chromatography–mass spectrometry (GC–MS) and selective ion flow tube–mass spectrometry (SIFT–MS) methods enable very accurate ammonia measurement. However, their use still has some drawbacks in terms of practicality, such as analysis by qualified staff, time-consuming sample preparation and measurement, and complicated operation and maintenance of equipment [39].

Humidity is one of the most influential sources of interference in the development of chemical sensors [40] and no practical method has been demonstrated to solve it yet. From this perspective, the sensor system demonstrated in this study offers a potential and feasible application of chemical sensors to detect ammonia in human breath and in industrial samples as well as ammonia mixed with other amine gases at almost saturated humidity levels. Furthermore, the current study provides a methodology for developing sensor architecture capable of utilizing a non-conventional photocrosslinking LBL method. Our future work will focus on improving the current sensor system by precisely investigating the contribution of hydrophobicity applied in the sensor films.

## Figures and Tables

**Figure 1 sensors-21-06176-f001:**
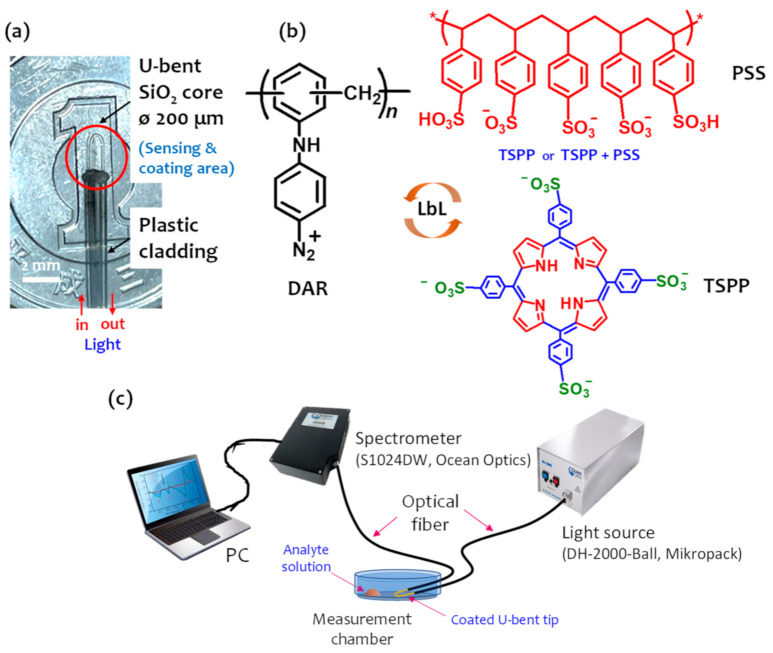
(**a**) Photograph of a U-bent optical fiber. (**b**) Chemical structures of DAR, TSPP, and PSS and a schematic of the LbL assembly of DAR and TSPP, or DAR and a TSPP+PSS binary mixture on a multimode U-bent optical fiber core followed by exposure to UV-radiation at 365 nm. (**c**) Schematic of the experimental setup for gas detection.

**Figure 2 sensors-21-06176-f002:**
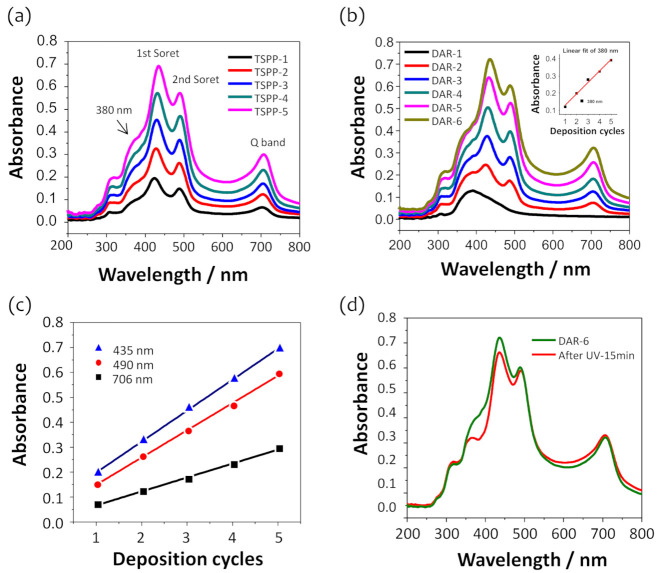
Evolution of the UV-vis absorption spectra of the DAR/TSPP+PSS (0.025 wt%) alternate layers deposited onto the 1-cm-long stripped core of a U-bent optical fiber when the outermost layer was deposited with (**a**) TSPP+PSS and (**b**) DAR, respectively. (**c**) Absorbance changes measured at (▲) 435, (●) 490, and (■) 706 nm. (**d**) Comparison of the UV-vis absorption spectra of the 5-cycle DAR/TSPP+PSS (0.025 wt%) film with an outermost layer of DAR before and after exposure to UV irradiation for 15 min.

**Figure 3 sensors-21-06176-f003:**
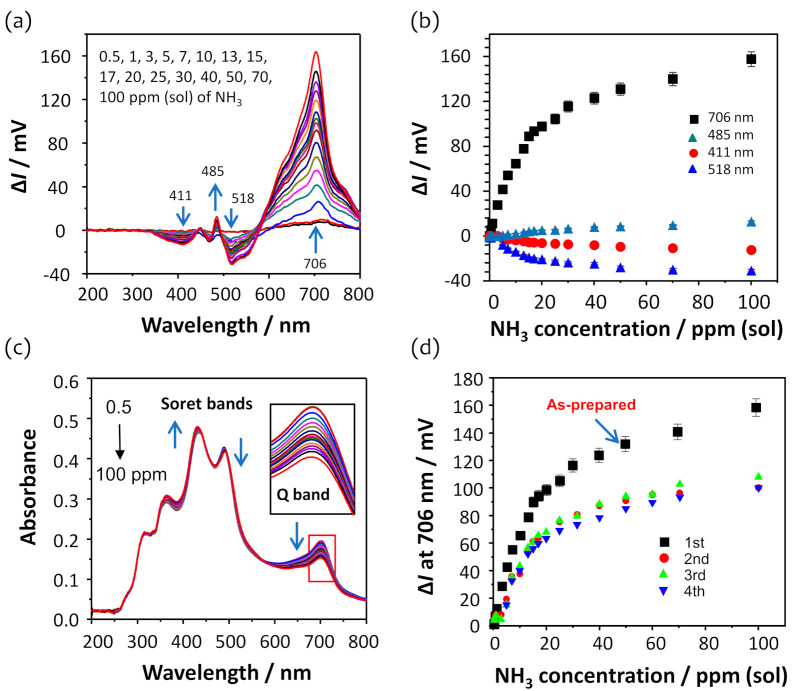
(**a**) Evolution of the transmission spectra induced by the exposure of the U-bent OFS modified with a 5-cycle DAR/TSPP+PSS (0.025 wt%) film to ammonia concentrations from 0 to 100 ppm (sol). (**b**) The intensity changes at four different wavelengths versus ammonia concentration, obtained from the three independent experiments conducted using the individual as-prepared sensors fabricated under the same conditions. (**c**) The corresponding UV-vis absorption spectral changes due to the exposure of the film to ammonia. The inset shows the decrease in absorbance of the Q band centered at 706 nm induced by the presence of ammonia. (**d**) The intensity changes at 706 nm, obtained from the four consecutive ammonia measurements.

**Figure 4 sensors-21-06176-f004:**
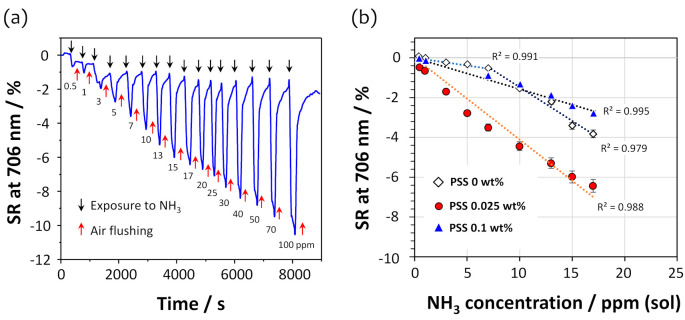
(**a**) Dynamic sensor responses (SRs) of the U-bent OFS coated with a 5-cycle DAR/TSPP+PSS (0.025 wt%) film at 706 nm when exposed to different ammonia concentrations from 0 to 100 ppm (sol). (**b**) SRs of the sensor films with different PSS contents of 0 wt%, 0.025 wt%, and 0.1 wt%, displaying a linear behavior within 17 ppm (sol) in each calibration curve.

**Figure 5 sensors-21-06176-f005:**
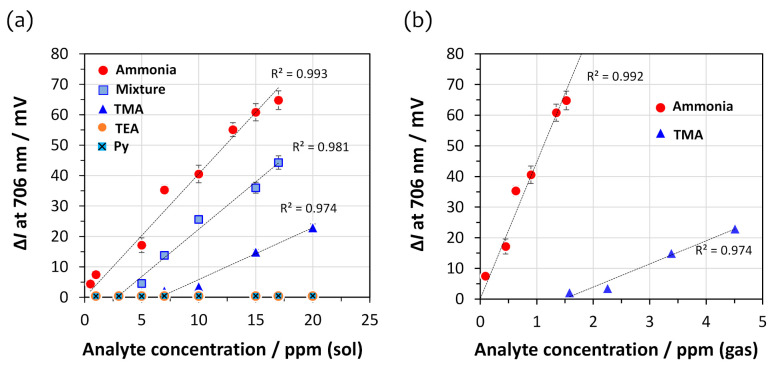
Calibration curves for the intensity change (Δ*I* as sensor response, *n* = 3) at 706 nm of the 5-cycle DAR/TSPP+PSS (0.025 wt%) film sensor (**a**) at different solution concentrations of ammonia, TMA, TEA, Py, and a mixture of all four analytes (equal concentrations in the range of 0–17 ppm for each), and (**b**) at different gas concentrations of ammonia and TMA.

**Figure 6 sensors-21-06176-f006:**
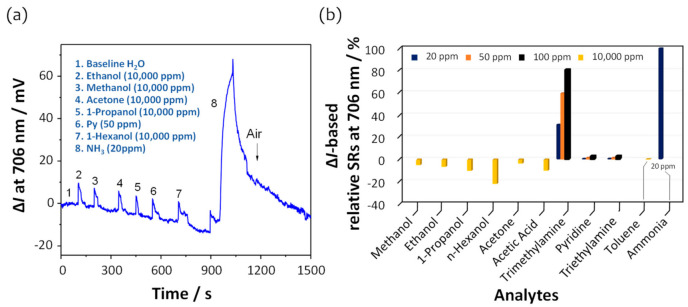
(**a**) Dynamic response to ammonia and other chemical analytes. (**b**) Comparison of the intensity change-based relative SRs at 706 nm on exposure to ammonia (20 ppm), TMA, TEA, Py (20 ppm, 50 ppm, 100 ppm, respectively), and all other analytes at 10,000 ppm.

**Figure 7 sensors-21-06176-f007:**
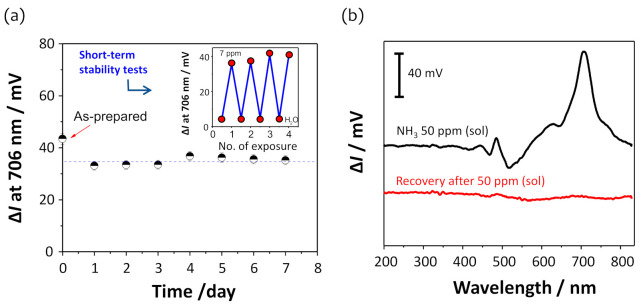
(**a**) Long-term stability of the sensor response due to the repeated exposure (*n* = 5) to 7 ppm (sol) ammonia over seven days. The inset shows repeatable switching of intensity changes obtained in a series of repeated ammonia exposure experiments in a short period of time. (**b**) Intensity changes in the transmission spectra due to the exposure of the U-bent OFS to 50 ppm (sol) ammonia and after flushing with dry air.

**Figure 8 sensors-21-06176-f008:**
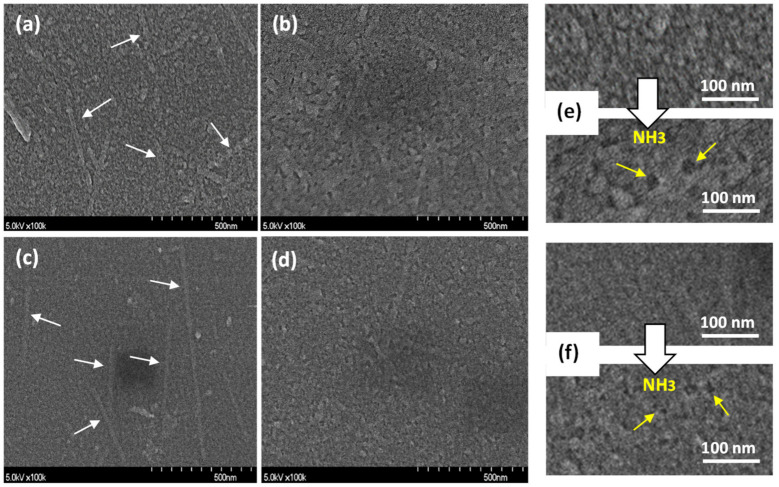
SEM images of (**a**,**b**) 5-cycle DAR/TSPP and (c and d) 5-cycle DAR/TSPP+PSS (0.025 wt%) films deposited on silicon wafer substrates (a and c) before exposure and (**b**,**d**) after exposure to 10,000 ppm (sol) ammonia gas for 2 min. The arrows in white in (**a**,**c**) point to TSPP J-aggregates. Magnified SEM images showing dozens of holes (see arrows in yellow in both cases) after ammonia gas exposure in the sensor films (**e**) without PSS and (**f**) with PSS (0.025 wt%).

**Figure 9 sensors-21-06176-f009:**
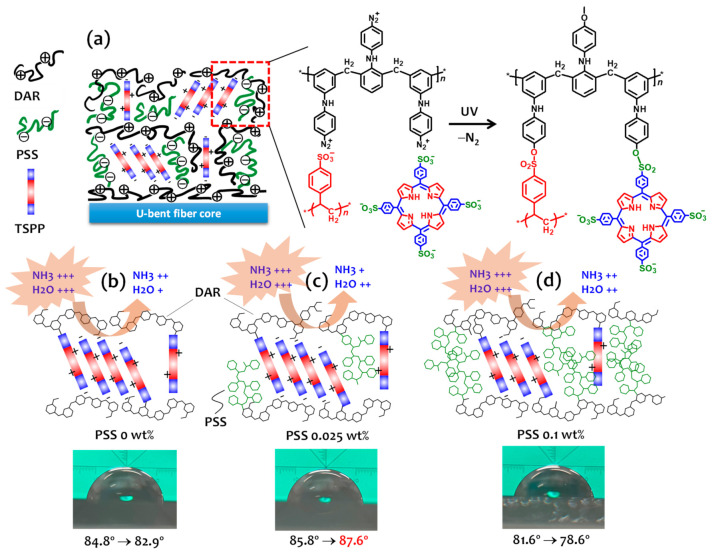
(**a**) Photochemical crosslinking reaction between DAR and the mixture of TSPP and PSS within the multilayered thin film. Schematic illustration of the DAR/TSPP+PSS films with (**b**) PSS 0 wt%, (**c**) PSS 0.025 wt%, and (**d**) PSS 0.1 wt% and photographs of contact angles of the corresponding films. The arrows indicate changes in the contact angle from DI water to a 70-ppm ammonia solution.

**Table 1 sensors-21-06176-t001:** Sensing parameters of the DAR/TSPP+PSS films with a different PSS content.

Film Name	Sensitivity (Slope) ^a^, % ppm^−1^ (*R*^2^)	Response Time ^b^, s	Recovery Time, s	Linear Range ^b^, ppm (sol)	LOD (sol) ^c^, ppm (sol)
DAR/TSPP+PSS (0 wt%)	0.08 (0.991) 0.34 (0.979)	100	180	0–7 7–17	1.25 2.81
DAR/TSPP+PSS (0.025 wt%)	0.41 (0.988)	30	30	0–17	0.23
DAR/TSPP+PSS (0.1 wt%)	0.16 (0.997)	45	40	0–20	0.61

^a^ Data measured at 120 s. ^b^ Response time determined as the interval (τ_90_) needed for the signal to achieve 90% of its saturation measured at 50 ppm ammonia (sol). ^c^ Limit of detection (LOD) was estimated to be 0.096% SR as a 3σ level, where σ is 0.032% SR as a possible noise value.

**Table 2 sensors-21-06176-t002:** Sensitivity and LOD of the DAR/TSPP+PSS (0.025 wt%) film obtained when the solution and gas concentrations for ammonia and TMA were used.

Analyte	Sensitivity (sol) ^a^, mV ppm^−1^ (R^2^)	LOD (sol) ^d^, ppm	Sensitivity (gas), mV ppm^−1^(R^2^)	LOD (gas) ^d^, ppm
Ammonia	4.05 (0.993) ^b^	0.31	44.7 (0.992)	2.85 × 10^−2^
TMA	1.73 (0.974) ^c^	7.21	7.56 (0.974)	1.65
Mixture	3.11 (0.974) ^c^	3.23	NA	NA

^a^ Data measured at 120 s. Applied concentration ranges: ^b^ 0–17 ppm and ^c^ 7–20 ppm. ^d^ LOD was estimated to be 1.28 mV as a 3σ level, where σ is 0.43 mV as a possible noise value (Appendix A). NA: not applicable.

**Table 3 sensors-21-06176-t003:** A summary based on the comparison of the proposed ammonia sensor to the previously published OFSs.

Sensing Platform	Sensing Material	Fabrication Method	LOD (ppm)	Response Time	Recovery Time	Humidity Range
U-bent optical fiber (this study)	DAR/TSPP+PSS	Crosslinked LbL	0.03	30 s (r.t.)	30 s (r.t.)	85%
Fiber tip [35]	TSPP	Sol–gel	0.15	83 s	NA	<70%
U-bent optical fiber [36]	Bromocresol purple	Dip coating sol–gel	10 (55.5 °C)	5 min (r.t.) 10 s (55.5 °C)	20 min (r.t.) 10 min (55.5 °C)	NA
Side polished fiber [37]	Graphene/polyaniline	Chemical in-situ polymerization	22.5	112 s	185 s	NA
Linear optical fiber [22]	TSPP/PDDA	LbL	3	96 s	192 s	NA
Fiber optic grating [28]	TSPP/PDDA	LbL	0.67	NA	NA	NA
Tapered fiber [38]	TSPP/PAH *	LbL	2	100 s	240 s	NA

NA: not applicable; r.t.: room temperature; * poly(allylamine hydrochloride).

## Data Availability

Not applicable.

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
