# Peer review of "Fabrication of Humidity-Resistant Optical Fiber Sensor for Ammonia Sensing Using Diazo Resin-Photocrosslinked Films with a Porphyrin-Polystyrene Binary Mixture"

_sensors, 2021, doi:10.3390/s21186176_

Round 1
Reviewer 1 Report
The authors of ‘Fabrication of Humidity-Resistant Optical Fiber Sensor for Ammonia Sensing using Diazo Resin-Photocrosslinked Films with a Porphyrin-Polystyrene Binary Mixture’ report a new colorimetric ammonia gas sensor utilizing a U-bent optical fiber modified with multiple layers of diazo resin (DAR), tetrakis(4-sulfophenyl)porphine (TSPP) and poly(styrene sulfonate) (PSS) that form a UV-sensitive polymeric sensing layer. The sensor performance characteristics were well studied and well explained resulting in a sensor with a linear range up to 17 ppm (solution) and a good selectivity. The manuscript can be published after addressing the following comments, which overall require minor changes:
- Can the authors improve the quality of Figure 1b and Figure 9a in order to clarify the structure and functional groups?
- Section 3.1 Strategy for sensitivity and reproducible sensor fabrication, could the authors include more details about how the sensitivity of the sensor was increased by the covalent bond-based crosslinking of the sensing layer approach? The effect of hydrophobicity is explained however, there is no clear link between the sensitivity and reproducibility of the sensor`s response and the fabrication method in this section.
- Figure 3b, is the data obtained with at least three independent experiments, e.g. with three sensors where all were used as prepared, or after being used for the same amount of time? The error bars are not visible.
- ‘This decrease in intensity change after the first measurement of the as-prepared film may be due to the optimization of the film structure after continuous use.’ Could the authors elaborate the optimization effect of repetitive uses?
- Figure 5 and 7, please note the number of experiments from which the data was acquired for the graph.
- Section 3.5 Sensitivity and selectivity of the PSS-containing sensor films, what is the intended area of use of the sensor and what are the implications of the presence of TMA on the reliability of the sensor`s response given that a significant interference can be observed for concentrations higher than 2 ppm of TMA?
- Section 3.6. Stability and reproducibility of the sensing system, could the authors quantify the reusability of the sensor?
- What are the advantages of using the explained ammonia sensor fabrication method over the reported ones? A comparison of the method and the sensor performance characteristics of the U-bent OFS coated with a DAR/TSPP+PSS with the reported sensors will be helpful in understanding this.
- What is/are the standard method/s used for gaseous ammonia detection (example: Berthelot methods-based colorimetric approaches that require the use of spectrophotometers) and could the authors compare the sensor`s performance with that standard method of choice?
Minor typos:
- ‘using’ word is not capitalized in the title sentence.
- Supplementary file: Page 6, Figure 9 instead of Figure S9.
- Line 45: ‘striped’ fiber.
- Line 79 and 276: ‘stabler’.
- Line 266: ‘measured four times’.
- Line 436: ‘a faster sensor response time’.
Author Response
Dear Reviewer,
Please find our enclosed manuscript entitled as follows:
Fabrication of Humidity-Resistant Optical Fiber Sensor for Ammonia Sensing Using Diazo Resin-Photocrosslinked Films with a Porphyrin-Polystyrene Binary Mixture
Manuscript ID: sensors-1249395
Authors: Soad Ahmed, Yeawon Park, Hirofumi Okuda, Shoichiro Ono, Sergiy Korposh, Seung-Woo Lee *
All comments / queries / suggestions from the reviewer have been considered to improve the manuscript quality. Changes have been highlighted in yellow in the revised manuscript.
We hope after the current improvement our work will be suitable for publication in Sensors.
Sincerely yours
Seung-Woo Lee, Ph. D, Prof. (leesw@kitakyu-u.ac.jp)

Reviewer 2 Report
Ahmed et al. report an optical fiber based ammonia gas sensor with enhanced sensing capability by introducing hydrophobic poly styrene sulfonate into the coating of fiber core. The resulting sensor showed good sensitivity, faster response time and less interference by humidity. The manuscript was well written and the authors provides sufficient evidence for their hypothesis. I would suggest accepting this manuscript after minor revision.
- Figure 1b needs to be reproduced with higher quality.
- The authors need to add the discussion on why they chose a vaporizing ammonia solution to control the gaseous ammonia concentration instead of ammonia gas in controlled humidity. The current set-up was kind of limited by a saturated water vapor pressure at specific temperature.
- In page 2, the authors mentioned the ammonia application as an energy vector. But the authors needs to correlate it to their detection technique. Are you using it for ammonia manufacturing, transportation or utilization.
- I am wondering how this ammonia detection technique can be used in aqueous ammonia solutions (not ammonia gas vaporized from ammonia solutions).
- The authors mentioned that the ammonia sensor could be fully recovered by flushing air for less than 30 sec. However, the flushing air provided a huge ammonia concentration gradient to remove the ammonia molecules. In reality, this is not the case. Could the authors suggest how long it may take if, for example, 100 ppm ammonia is replaced by 20 ppm ammonia?
Author Response

(The authors gave the same response as above.)
